# Effect of Methyl Jasmonate on Phenolic Accumulation in Wounded Broccoli

**DOI:** 10.3390/molecules24193537

**Published:** 2019-09-30

**Authors:** Yuge Guan, Wenzhong Hu, Aili Jiang, Yongping Xu, Rengaowa Sa, Ke Feng, Manru Zhao, Jiaoxue Yu, Yaru Ji, Mengyang Hou, Xiaozhe Yang

**Affiliations:** 1School of Bioengineering, Dalian University of Technology, Dalian 116024, China; guanyuge@foxmail.com (Y.G.); xyping@dlut.edu.cn (Y.X.); kuailexiaosa@sina.com (R.S.); jyr2011037110@mail.dlut.edu.cn (Y.J.); mengyanghou@yahoo.com (M.H.); yxiaozhe@163.com (X.Y.); 2College of Life Science, Dalian Minzu University, Dalian 116600, China; jal@dlnu.edu.cn (A.J.); fengkessky@sina.com (K.F.); ZhManRu@163.com (M.Z.); huaqianshua@163.com (J.Y.); 3Key Laboratory of Biotechnology and Bioresources Utilization, Ministry of Education, Dalian 116600, China

**Keywords:** wounded broccoli, methyl jasmonate, phenolic accumulation, phenylpropanoid metabolism, antioxidant capacity

## Abstract

In order to find an efficient way for broccoli to increase the phenolic content, this study intended primarily to elucidate the effect of methyl jasmonate (MeJA) treatment on the phenolic accumulation in broccoli. The optimum concentration of MeJA was studied first, and 10 μM MeJA was chosen as the most effective concentration to improve the phenolic content in wounded broccoli. Furthermore, in order to elucidate the effect of methyl jasmonate (MeJA) treatment on phenolic biosynthesis in broccoli, the key enzyme activities of phenylpropanoid metabolism, the total phenolic content (TPC), individual phenolic compounds (PC), antioxidant activity (AOX) and antioxidant metabolism-associated enzyme activities were investigated. Results show that MeJA treatment stimulated phenylalanine ammonia-lyase (PAL), cinnamate 4-hydroxylase (C4H), and 4-coumarin coenzyme A ligase (4CL) enzymes activities in phenylpropanoid metabolism, and inhibited the activity of polyphenol oxidase (PPO), and further accelerated the accumulation of the wound-induced rutin, caffeic acid, and cinnamic acid accumulation, which contributed to the result of the total phenolic content increasing by 34.8% and ferric reducing antioxidant power increasing by 154.9% in broccoli. These results demonstrate that MeJA in combination with wounding stress can induce phenylpropanoid metabolism for the wound-induced phenolic accumulation in broccoli.

## 1. Introduction

Broccoli is a well known green vegetable with great commercial value owing to its special nutritional and functional components such as phenolic compounds (quercetin, caffeic, quinic acid, chlorogenic acid, etc.) [1,2], vitamin C (Vit C), and other bioactive compounds (Zn, fiber, glucosinolates, etc.) [3,4]. Presently, fresh-cut broccoli has become more popular due to its nutritional benefits and convenience properties [5]. During fresh-cut processing, the tissue is inevitably subjected to wounding stress, which will induce defense responses to produce more secondary metabolites at the injured site or site adjacent to defend and heal the wounding damage [6]. It has been confirmed that wounding stress induces phenolic biosynthesis and enhances antioxidant capacity in various fruit and vegetables, such as carrot [7], potato [8], onion [9], pitaya [10], lettuce [11], celery [11], and sweet potato [11]. These findings indicate that wounding can be used as an effective and practical means to enhance the phenolic accumulation and improve the nutritional quality of the postharvest product.

As one of the most important secondary metabolites, phenolic compounds are synthesized through the phenylpropanoid metabolic pathway in fresh-cut produce. In the phenylpropanoid metabolic pathway, PAL, C4H, and 4CL are three key enzymes that catalyze the synthesis of phenolic compounds in plants [9]. It was reported that wounding stress induced high PAL enzyme activity resulting in the phenols accumulation in carrot [12] while the high activity of PPO enzyme decreased the phenolic content by enhancing phenol oxidation in apple [13]. Therefore, the accumulation of phenols depends on the dynamic change between phenol synthesis and oxidation reaction. Furthermore, the level of phenols was regulated by many postharvest abiotic stresses, such as MeJA [8,14], ethylene, salicylic acid, systemin [15,16,17], UV radiation [18], citric acid [19], and light exposure [20] in wounded fruit and vegetables.

MeJA is an important plant hormone that has crucial roles in plant resistance, as well as an essential signaling molecule participating in defense reactions responding to abiotic and biotic stress [8,21]. It has confirmed that 1 μmol L^−1^ MeJA could effectively increase the level of total phenols induced by chilling stress in fresh peach [17]. Research on wounding stress also showed that 0.025% MeJA treatment significantly stimulated the wound-induced biosynthesis of phenols in carrots and lettuces [22] by increasing in PAL activity. Similarly, it was reported that MeJA in combination with wounding stress play crucial roles in modulating the ROS level and activating secondary metabolisms to induce higher phenolic accumulation in wounded pitaya [23], purple-flesh potatoes [24], celery and nectarine [22]. However, it is still unclear how MeJA regulate the phenolic content of broccoli. Therefore, the phenolic compounds content, the key enzymes that catalyze the synthesis of phenolic compounds and the antioxidant capacity were investigated in wounded broccoli after treatment with MeJA, in order to evaluate the effect of MeJA treatment on the phenolic accumulation in wounded broccoli.

## 2. Results

### 2.1. Effect Treatments with Different MeJA Concentrations on the Total Phenolic Content (TPC) and Ferric Reducing Antioxidant Power (FRAP)

As shown in Figure 1a, the changes of TPC of all samples increased first and then decreased with the extension of storage time. At 36 h, the maximum value of TPC was obtained in 10 μM MeJA treatment, which increased 34.8%, 16.5% compared to control group at 0 h and 36 h, respectively. Similar to the trending of TPC, the FRAP (Figure 1b) attained its maximum at 36 h and 10 μM MeJA treatment group showed 2.05-fold higher than the whole broccoli before storage. However, high concentration of MeJA treatment reduces the TPC and FRAP, while the effect of 1 μmol L^−1^ MeJA treatment on TPC accumulation is not obvious. Therefore, the optimum concentration of MeJA was 10 μM, and this concentration was used later in the following experiments to study the role of MeJA for the effect on the phenolic accumulation in wounded broccoli.

### 2.2. Effect of MeJA Treatment and Wounding Stress on the Total Phenolic Content (TPC) and Antioxidant Capacity (AOX)

According to Figure 2, TPC (a), FRAP (b), DPPH (c), and ABTS (d) radical scavenging capacity in each treatment group changed slightly at the first 12 h of storage and the parameters of the two wounding stress groups showed rapid upward trend subsequently at 24 and 36 h while without wounding stress groups showed a slow downward trend. At 36 h, TPC, FRAP, DPPH and ABTS of MeJA + wounded group increased by 34.8%, 154.9%, 55.6%, 116.3%, respectively, compared with control, which was significantly higher than wounded broccoli (*p* < 0.05). Similarly, MeJA treatment could effectively delay the TPC and AOX decline in the whole groups. At 12, 24, 36, 48 h, the TPC of MeJA group was 1.2%, 2.4%, 4.5%, 3.6% higher than control, respectively. The AOX of MeJA group was 1.6–24.1% higher than control during the whole storage. On the other hand, the wounding stress induced the phenolic accumulation and increased the AOX. At 36 h, the TPC, FRAP, DPPH, and ABTS of the wounded group was higher 16.7%, 150%, 36.7%, 98.2%, respectively compared with control. Correlation analyses showed that there was a significant correlation between TPC and FRAP, DPPH radical scavenging capacity, ABTS radical scavenging capacity, and the correlation coefficients were 0.923, 0.848, 0.891, respectively.

According to Figure 3, the FRAR value of 40 μg mL^−1^ ascorbic acid was 16.80 µmol Trolox mL^−1^, and the DPPH and ABTS radical scavenging activity of IC_50_ values were 17.77 and 55.12 μg mL^−1^, respectively. Hence, the total phenolic extract from broccoli was considered to have a high antioxidant capacity.

### 2.3. Comparative Analysis of Individual Phenolic Compounds by HPLC-PAD.

Phenolic compounds from broccoli have important biological activities, therefore, qualitative and quantitative analysis of these phenolic compounds in broccoli will be of great significance for quality evaluation and bioactivity study. The HPLC-PAD chromatograms of phenolic standard and individual phenolic compounds from broccoli as shown in Figure 4. According to Table 1, among the different changes observed, the content of caffeic acid, sinapic acid, cinnamic acid, showed a gradual increase and catechin, hydroxybenzoic acid, chlorogenic acid, quercetin, ferulic acid decreased obviously (*p* < 0.05) in the wounded group. At the end of storage, the contents of caffeic acid, sinapic acid, cinnamic acid and quercetin in the wounded group were significantly higher than control. At 48 h, the contents of chlorogenic acid, rutin, caffeic acid, and cinnamic acid in MeJA + wounded group were increased by 1.99, 1.77, 1.85, and 31.59 times compared to the wounded group, respectively. Moreover, the individual phenolic compounds in control gradually decreased during the whole storage, however, the contents of sinapic acid, ferulic acid, cinnamic acid, chlorogenic acid, and quercetin in MeJA group were obviously (*p* < 0.05) higher than control, this result indicated that MeJA treatment significantly delayed the phenol decline in the whole broccoli.

### 2.4. Effect of MeJA Treatment and Wounding Stress on PAL, C4H, and 4CL Activity

Phenols are the main secondary metabolites in plants and are mainly synthesized through the phenylpropanoid metabolism and PAL, C4H, and 4CL enzymes are the key enzymes of this pathway. According to Figure 5a, PAL activity was continuously increased in wounded broccoli before 36 h, which was 5.72 times higher than the whole broccoli before wounding. On the contrary, PAL activity in whole broccoli substantially reduced during all types of storage. MeJA treatment further significantly (*p* < 0.05) accelerated PAL activity and it was 1.79 times higher than wounded broccoli at the end of storage but had no obvious effect on PAL activity level in comparison with whole broccoli.

As shown in Figure 5b,c, the activity of C4H and 4CL in wounded broccoli increased markedly during the first 24 h, then experienced a relatively stable trend afterward. Similarly, as the PAL activity, MeJA treatment has a significant (*p* < 0.05) effect on C4H and 4CL activities and maintains 35.1% and 16.1% higher level than wounded broccoli at 24 h during storage. For the whole broccoli, the activities of C4H and 4CL declined slowly in all types of storage. However, MeJA treatment kept a high level and there was a 1.35 and 1.11 times increase of C4H and 4CL activities compared with control at the end of storage, respectively.

### 2.5. Effect of MeJA Treatment and Wounding Stress on Vit C Content and APX, PPO, and POD Activity

As shown in Figure 6a, broccoli was rich in Vit C and the initial content was 77.4 mg kg^−1^. As the storage time progressed, Vit C content in each group decreased gradually and wounding stress group with a sharp downward trend, whereas, MeJA treatment significantly restrained the decrease of the decline of Vit C content and the value of this parameter was 8.9% higher than wounded broccoli at 24 h. In contrast, the Vit C content of control and MeJA group was slowly decreasing as storage time progressed and there was no significant difference between the two groups.

During the whole storage, APX activity decreased gradually as storage time progressed and there was 11.4% and 42.3% decrease in control and wounded broccoli, respectively (Figure 6b). MeJA treatment significantly delayed the wounding stress-induced decline of APX activity, and APX activity in MeJA + wounding broccoli was 288.6 U g^−1^ at 36 h, showing 1.37 times higher than that of the wounded broccoli, whereas, there was no obvious difference (*p* > 0.05) between groups without wounding stress.

PPO activity in all groups just increased during 12–24 h and then continued to slow down until relatively stable at the end of storage. MeJA treatment significantly inhibited PPO activity and the values of this parameter in MeJA and MeJA + wounding broccoli were 0.89 U g^−1^ and 0.67 U g^−1^, which was 1.23 times and 1.40 times lower than that without MeJA treatment, respectively, at 48 h during storage (Figure 6c).

POD activity in whole broccoli increased during the 12–24 h of storage and then gradually reduced in other storage time, while MeJA treatment significantly (*p* < 0.05) suspended the storage time-induced reduction and maintained higher levels of them in whole broccoli after 24 h during storage. Wounded broccoli showed lower POD activity in comparison with the intact groups (Figure 6d).

## 3. Discussion

As an important antioxidant, phenols act as promoters of human health through their scavenging activity by preventing chronic diseases such as cardiovascular diseases, cancers, type 2 diabetes, neurodegenerative diseases [25,26]. Therefore, it is necessary to find an effective technology that can ensure the delivery of products with high levels of the desired antioxidants. The present study shows that the phenolic content of wounded broccoli was markedly affected by different MeJA concentrations, and 10 μM MeJA had a significant influence on phenolic accumulation (*p* < 0.05). Similar optimum treatment concentration of MeJA was also found in Chinese bayberries [27]. In other studies, the effective concentration of MeJA on promoting antioxidant capacity ranged from 1 to 1000 μM [28,29]. This result may be caused by different treatment methods and the type of fruits and vegetables. Our studies showed that low concentration (1 μM) MeJA had little effect on TPC in broccoli during all the storage period, which is probably because the concentration of MeJA was too low to induce the mechanism of resistance to wounding stress. When the concentration of MeJA was greater than 100 μM, MeJA treatment had a slight inhibitory effect on TPC in broccoli, this result is in agreement with a previous report, where the application of 250 ppm MeJA to wounded broccoli for 24 h at 20 °C did not induce a significant increase in the concentration of total phenolics [14].

Fruits and vegetables contain diverse phenolic compounds, including cinnamic acid, gallic acid, caffeic acid, chlorogenic acid, catechol, epicatechol, guaiacol and its polymers and esters, and they can also be induced to synthesize by mechanical injury during the processing of fresh-cut fruit and vegetables [29,30], which make great contribution to antioxidant activity and further improve the health properties [31,32]. The content of caffeic and sinapic acid increases 4.35, and 5.87 times at 36 h, which contributes to the enhancement of antioxidant activity in wounded broccoli. Moreover, the accumulation of phenols could scavenge free reactive oxygen radicals and inhibit membrane lipid peroxidation, thereby increasing the plant’s resistance to oxidative damage and further preservation of the product [33]. Sinapic acid is one of the principal precursors of lignin [34]. Therefore, the higher levels of sinapic acid observed after storage of wounded broccoli may be related with the wound-induced activation of the phenylpropanoid metabolism, which is required for the biosynthesis of lignin that, in wounded plant tissue, serves as a water-impermeable barrier that prevents excessive water loss and improves the resistance [34,35]. In order to illustrate the relationship among individual phenols, the total phenolic content and antioxidant activity, a principal component analysis (PCA) was performed, as shown in Figure 7. According to the PCA results, there were two principal components (PC) among these parameters. The PC1 mainly includes total phenols, antioxidant activity, and five kinds of individual phenols (rutin, quercetin, cinnamic acid, caffeic acid, sinapic acid). This result indicates that rutin, quercetin, cinnamic acid, caffeic acid, and sinapic acid contribute to the enhancement of the total phenolic content and antioxidant activity of wounded broccoli. Among the five kinds of individual phenols in PC1, cinnamic acid, caffeic acid, and sinapic acid were closer to the total phenolic content and antioxidant activity in the chart of PCA. It means that these individual phenols have a higher contribution to the total phenolic content and antioxidant activity. The PC2 mainly includes chlorogenic acid, ferulic acid, hydroxybenzoic acid, and catechin. This result suggests that the four kinds of individual phenols had a lower contribution to the increase of the total phenolic content and antioxidant activity.

The application of MeJA to wounded broccoli increased the accumulation of chlorogenic acid, which was similar to the previous report for the 9.0 kJ m^−2^ UV-C treatments on fresh-cut broccoli [18]. Furthermore, MeJA treatment could also induce the accumulation of rutin, caffeic, cinnamic, and ferulic acid, which contributes to increasing the TPC in wounded broccoli 1.36 times at 36 h compared to the initial value. However, Villarreal-García [14] reported that the TSP content showed no significant change during 24 h storage at 20 °C in broccoli with MeJA treatment. What is different in this study, compared with previous reports, is that MeJA is applied prior to wounding, whereas in the other studies, MeJA is applied during storage of wounded tissue. The present study has shown that the TPC in MeJA, wounding, MeJA + wounded group increased by 4.4%, 16.5%, 36.0% respectively, compared to the control group before storage. As shown in Figure 8, the total amount of the individual phenolic content in MeJA, wounding, MeJA + wounded group increased by 5.2%, 295.4%, 455.5%, respectively, at 36 h, compared to the control group. According to these results, no matter TPC or the total amount of nine kinds of the individual phenolic content, the phenolic accumulation of MeJA treatment on the wounded group was evidently higher than the whole broccoli. These findings concluded that MeJA, in combination with wounding stress, can synergistically induce the phenolic accumulation in wounded broccoli.

Phenols are the main secondary metabolites in plants which are mainly synthesized through the phenylpropanoid metabolism shown in Figure 9. PAL enzyme is the key enzyme of the initial step of phenylpropanoid metabolism and catalyzes l-phenylalanine to produce trans-cinnamic acid, which is the connection between primary metabolism and phenylpropanoid metabolism pathway. C4H catalyzes the conversion of trans-cinnamic acid to *p*-coumaric acid, and then acylated to *p*-coumaroyl CoA in the presence of C4H. Our study is the first report that indicates MeJA treatment with wounding stress induced the activities of PAL, C4H, and 4CL, which contributed to the accumulation of phenols in wounded broccoli, and similar results have been confirmed in wounded Chinese bayberries [27], potato [36], locuts root [37], carrot [12], and pitaya [10]. In addition, similar to the PAL activity, the tendency of C4H and 4CL matched with the accumulation of TPC and AOX. However, the effects of enzymes on the regulation of nine kinds of individual PC in broccoli are not entirely consistent. According to Table 2, the correlation analysis showed a significant positive correlation between the three major enzymes of phenylpropanoid metabolism and four kinds of individual phenols (caffeic acid, sinapic acid, quercetin, and cinnamic acid), whereas there was no significant correlation between the three enzymes and the other five kinds of individual phenols. Some factors contribute to this result, for example, some phenols such as ferulic acid, not only can be synthesized but also decomposed, and the synthesis mechanism of some individual phenols such as hydroxybenzoic acid and rutin are unclear [14]. Through the above study, we can conclude that the synthesis of phenolic compounds is a complex process, which is regulated by the interaction of various substrates and enzyme activities.

In order to maintain a high AOX, wounded fruit and vegetables have evolved mechanisms to protect membrane lipid against oxidation by antioxidant enzymes, such as APX, PPO, and POD. APX, as the major enzyme of reducing lipid peroxidation, catalyzes the degradation of H_2_O_2_ to H_2_O and O_2_. MeJA treatment significantly maintained a high APX activity (Figure 6) in the present study, which could be a primary reason contributing to the high level of Vit C. PPO and POD were the downstream enzymes that oxidize phenolic compounds to brown quinone. According to the correlation analysis, there was a significant negative correlation between PPO and TP, however, there is no significant correlation between POD and TP. This result indicates that the accumulation of TP was significantly affected by PPO, but not by POD, in broccoli. On the other hand, according to the correlation analysis between PPO and PCs, it is speculated that the optimum substrate of PPO in broccoli was sinapic acid, followed by caffeic acid, which was not consistent with the reports for apple [38] and litchi pericarp [39]. The best substrates of PPO in various products are different, which is the main reason for the inconsistent change rules of phenolic substances in plants [38]. Treated with MeJA, broccoli could significantly inhibit the activity of PPO, thereby reducing the degradation rate of individual phenols, which contributed to maintaining a high level of PC and further increase the antioxidant activity.

## 4. Materials and Methods

### 4.1. Chemicals and Reagents

Ethanol, sodium carbonate (Na_2_CO_3_), Disodium hydrogenorthophosphate, sodium dihydrogen phosphate, 30% hydrogen peroxide, Vit C, catechol, 2-methoxyphenol, β-mercaptoethanol, polyvinylpyrrolidone (PVPP), glycerol, tris-base, hydrochloric acid, and magnesium chloride (MgCl_2_) were obtained from Tianjin Kemiou Chemical Reagent Co., Ltd. (Tianjin, China). Folin–Ciocalteu reagent, potassium persulfate, leupeptin, phenylmethylsulfonyl fluoride, boracic acid, borax, and l-phenylalanine were purchased from Beijing Solarbio Science and Technology CO., Ltd. (Beijing, China). Gallic acid, hydroxybenzoic acid, chlorogenic acid, caffeic acid, sinapic acid, ferulic acid, rutin, cinnamic acid, catechin, and quercetin, were obtained from Shanghai Yuanye Biochemical Co., Ltd. (Shanghai, China). 2,2-diphenyl-1-picrylhydrazyl (DPPH), 2,2-Azinobis-3-ethylbenzthiazoline-6-sulphonate (ABTS) and adenosine triphosphate were obtained from Shanghai Aladdin Biochemical Co., Ltd. (Shanghai, China). Methyl jasmonate and methanol were obtained from Sigma Chemical Co. (St. Louis, MO, USA). Methanol was reagent of HPLC-grade. The other chemical reagents were of reagent grade.

### 4.2. Plant Material Preparation and Treatments

Broccoli (*Brassica oleracea* L.var. italic Planch) was purchased from local producers (Dalian, China) and transported to the laboratory within 2 h of harvest. After selection, broccoli was sterilized in 200 μL L^−1^ of sodium hypochlorite and washed with distilled water, then, it was put in a ventilator to dry naturally for 2 h. In the first experiment, the samples were treated with 0, 1, 10, 100, 250, 500 μmol L^−1^ (μM) MeJA vapor in 34 L closed containers at 20 °C for 12 h, respectively. After MeJA treatment, the vegetables were left in air at 20 °C for 2 h before the wounding process. Then broccoli was minimally processed with a flower head diameter of 5 cm, stem length of 5 cm, with uniform size and no damage or yellowing of flowers. Samples of about 60 g of broccoli per treatment were randomly packaged in a 20 × 10 × 4 cm polypropylene container and stored for 0, 12, 24, 36, and 48 h at 20 °C while monitoring the total phenolic content and total antioxidant capacity. The optimum concentration of MeJA was determined according to the first experiment.

In the second experiment, the two groups of vegetable were treated with 0 and 10 μM MeJA (based on the first experiment) vapor in 34 L closed containers at 20 °C for 12 h, respectively. In addition to grouping, the preprocessing method was the same as the first experiment. One subgroup was cut into a flower head diameter of 5 cm, stem length of 5 cm by a sharp stainless steel knife, the other subgroup was left intact without any wounding stress. Hence, the entire experiment included the following four treatments, as shown in Table 2: (1) Control, broccoli treated with neither cutting nor MeJA, (2) MeJA, broccoli treated with MeJA alone, (3) wounding, broccoli treated with cutting alone, also known as wounded broccoli, (4) MeJA + wounding, broccoli pretreated with MeJA and then subjected to cutting. All four treated groups of broccolis were stored at 20 °C for 48 h. Broccoli samples were collected and frozen with liquid nitrogen every 12 h, then smashed with a frozen crusher (M20, Masuko, Japan) for analysis of individual PC, total phenol and Vit C contents, AOX, phenylpropanoid metabolism-related enzymes, and antioxidant enzymes assays.

### 4.3. Total Phenols and Vitamin C (Vit C) Content Assay 

Five grams of frozen tissue samples were mixed with 20 mL of 80% ethanol and homogenized until reaching uniform consistency using digital homogenizer (T25, Guangzhou Guangpeng, China) to obtain 0.25 g mL^−1^ ethanol extract. The obtained mixtures were preserved in covered centrifuge tubes for ultrasound extraction for 40 min in darkness at 40 °C, subsequently, centrifuged at 12,000× *g* for 20 min and the TPC of the supernatant was determined using a modification of the method described by Hu [16]. One milliliter of 0.25 g mL^−1^ supernatant was added to 1 mL of Folin–Ciocalteu reagent. After that, 10 mL of 7.5% (*w*/*v*) Na_2_CO_3_ solution and 13 mL of distilled water was added, and the mixtures were incubated at 25 °C for 90 min before measuring at 765 nm. TPC was expressed as mg kg^−1^ of gallic acid equivalents on a fresh weight tissue basis, based on a standard curve prepared with a standard gallic acid solution.

The Vit C is determined as reduced Vit C in this study, and the determination of Vit C content was according to the method of Bessey [40]. The frozen broccoli powder (5 g) were added and 20 mL of 2% oxalic acid solution was mixed until well combined and then filtered. Five milliliters of supernatant was titrated with a calibrated 2,6-dichlorophenol indophenol solution until achieving pink color (30 s not change). Vit C content was expressed as mg kg^−1^ on a fresh weight tissue basis.

### 4.4. Individual Phenolic Compounds Assay

Individual phenolic compounds were analyzed according to the method reported by Becerra-Moreno [41] with minor modifications. Briefly, 20 μL of ethanol extract was injected in the HPLC system (1260 series, Agilent Technologies, Santa Clara, CA, USA). Chromatographic separations were performed on C_18_ reverse phase column (250 × 4.6 mm, 5 μm) using 1% formic acid-water (A) and methanol (B) as mobile phase at the constant flow rate of 0.8 mL/min. The following mobile phase gradient was used: 0–5 min, 5–10% B; 5.01–15 min, 10–20% B; 15.01–25 min, 20%–30% B; 25.01–35 min, 30% B; 35.01–45 min, 30%–50% B; 45.01–55 min, 50%–60% B and 55.01–60 min, 60%–5% B. Moreover, the polyphenol determination was analyzed by an Agilent HPLC system interfaced UV detector (λ = 280 nm) at 25 °C and identified by comparing their experimental retention times and standard curve with authentic standards. In this experiment, we analyzed nine kinds of phenols, including catechin, hydroxybenzoic acid, chlorogenic acid, caffeic acid, sinapic acid, ferulic acid, rutin, cinnamic acid, and quercetin, and the individual phenolic content was determined from a standard curve prepared with an individual phenol solution and was expressed as mg kg^−1^ on a fresh weight tissue basis. The standard curves of catechin, hydroxybenzoic acid, chlorogenic acid, caffeic acid, sinapic acid, ferulic acid, rutin, cinnamic acid, and quercetin were y = 15670x + 4058 (R^2^ = 0.9998), y = 43042x + 19,721 (R^2^ = 0.9997), y = 41024x + 611.36 (R^2^ = 0.9998), y = 7511.4x + 1641.4 (R^2^ = 0.9965), y = 76156x – 214,369 (R^2^ = 0.9629), y = 79743x + 28,630 (R^2^ = 0.9997), y = 21598x + 4702.6 (R^2^ = 0.9998), y = 237973x + 75,549 (R^2^ = 0.9998), y = 38634x + 2925.6 (R^2^ = 0.9999), respectively, the range of these standard curves was 0.1 to 100 μg mL^−1^.

### 4.5. Antioxidant Capacity (AOX) Assay

The AOX was analyzed through the methods of ferric reducing antioxidant power (FRAP), DPPH and ABTS free radical scavenging according to the procedure of Chen [42] with some modifications. The extraction method was the same as the extracts prepared for the TPC assay, four times dilution of ethanol extraction solution for detection of antioxidant capacity.

The FRAP reagent was prepared according to Chen [42]. Six microliters of extract supernatant was mixed with 180 μL FRAP reagent and 18 μL double-steamed water, the reaction system was incubated at 20 °C for 20 min and performed spectrophotometrically (Hitachi U-2800 Spectrophotometer, Tokyo, Japan) at 593 nm. The results are expressed as µmol g^−1^ Trolox equivalents per fresh weight mass of tissue based on a calibration curve of Trolox (ranging from 1 µM to 20 µM final concentration).

Ethanol extract supernatant (0.1 mL; 0.0625 g mL^−1^) was added into 0.12 mM of DPPH solution (0.1 mL) and then mixed for 10 s and incubated in darkness for 30 min at 20 °C. Ethanol instead of supernatant was used as control and the absorbance at 517 nm was measured (Hitachi U-2800 Spectrophotometer, Japan). Results were calculated with the following formula according to Li [21]: DPPH radical scavenging activity (%) = [(A_0_− A_1_)/A_0_] × 100(1)
where A_0_ and A_1_ are the absorbance of the control and the sample, respectively.

The ABTS reagent was prepared with 2.45 mM of potassium persulfate and 7 mM ABTS and adjusting the absorbance of the solution to 0.7 with ethanol at 734 nm. The extract supernatant (20 μL) was added into ABTS solution (20 μL) and then mixed for 15 s and incubated in darkness for 6 min at 20 °C. Ethanol (80%), instead of supernatant, was used as a blank and ethanol was used for baseline correction. The evaluation of this parameter is similar to FRAP and expressed as µmol Trolox equivalents per gram (µmol Trolox g^−1^ ) of fresh weight tissue based on a calibration curve of Trolox (ranging from 1 to 20 µM of final concentration). Ascorbic acid (Vc) was used as a positive control in antioxidant capacity (FRAR, DPPH, ABTS) assay and the IC_50_ was defined as the concentration of ascorbic acid that resulted in a 50% inhibition of DPPH and ABTS radical.

### 4.6. Phenylpropanoid Metabolism-Related Enzymes Assays

PAL activity was measured according to the method of Heredia [22] with minor changes. Frozen tissue sample powder (2.5 g) was mixed with 10 mL of ice-cold borate buffer (50 mM, pH 8.5), then the enzyme extracts were centrifuged at 12,000× *g* for 30 min at 4 °C which prepared for enzyme activity analysis. The PAL reaction system was prepared with 3 mL of 50 mM of borate buffer, 0.5 mL of 20 mmol L^−1^
l-phenylalanine, 0.5 mL of the enzyme extraction solution and the absorbance at 290 nm was determined immediately (OD_0_). The mixture was incubated at 37 °C for 60 min and the reaction was stopped by adding 0.1 mL of 6 mol L^−1^ HCl, then determined absorbance (OD_1_) subsequently. One unit of U (enzyme activity) was equal to a change of 0.01 at 290 nm per min, and PAL activity expressed as units per kilogram of fresh weight, U kg^−1^, where U = 0.01 ΔA_290 nm_ per min.

The pH 8.9, 50 mmol L^−1^ Tris-HCl buffer is composed of 10 μmol L^−1^ leupeptin, 1 mmol L^−1^ phenylmethylsulfonyl fluoride, 5 mmol L^−1^ ascorbic acid, 15 mmol L^−1^ β-mercaptoethanol, 0.15% (*w*/*v*) PVPP, 4 mmol L^−1^ MgCl_2_ and 10% glycerol, which was the extracting agent of 4CL and C4H. One gram of frozen tissue sample powder was homogenized with 5 mL ice-cold Tris-HCl buffer and centrifuged at 12,000× *g* for 30 min at 4 °C, which prepared for 4CL and C4H activities analysis. The determination of 4CL activity was assayed as described by Knobloch [43], which was measured at 333 nm and expressed as U kg^−1^ on a fresh weight basis, where U = 0.01 ΔA_333 nm_ per min.

The reaction mixture of C4H activity was prepared according to the method of Han [12], which was measured at 340 nm immediately. Then incubated at 25 °C for 30 min, the mixture was stopped by adding 100 μL of 6 mol L^−1^ HCl and the absorbance was read at 340 nm. C4H activity was expressed as U kg^−1^ on a fresh weight basis, where U = 0.01 ΔA_340 nm_ per min.

### 4.7. Antioxidant Enzymes Assays

Two grams of frozen tissue powder was mixed with 10 mL ice-cold extraction buffers and then centrifuged at 12,000× *g* for 30 min at 4 °C, which prepared for PPO, POD, and APX activity analysis. For PPO and POD, the extraction buffer was 0.2 mol L^−1^ sodium phosphate buffer (pH 6.4) containing 4% PVPP, while the buffer of APX containing 1 mM ascorbic acid, 0.1 mM EDTA and 1% PVPP.

PPO activity was assayed as described by Tian [44] and POD activity was evaluated according to the method of Chen [13]. PPO and POD activities were expressed on a fresh weight basis as U kg^−1^, where U _PPO_ = 0.01 ΔA_398 nm_ per min, U _POD_ = 0.01 ΔA_460 nm_ per min.

APX activity was analyzed based on Nakano and Asada [45] with minor modifications. The reaction system of 3 mL was constructed by adding 2.6 mL sodium phosphate buffer (50 mM, pH 7.5), 0.1 mL enzyme extract and 0.3 mL H_2_O_2_ (2 mM), in turn. APX activity was expressed on a fresh weight basis as U kg^−1^, where U = 0.01 ΔA_290 nm_ per min.

### 4.8. Statistical Analysis

The experimental results were expressed as mean ±SD (standard deviation) of triplicate measurements and subjected to statistical analysis with the software of IBM SPSS Statistics 20. The principal component analysis was determined by IBM SPSS Statistics 20. All data were analyzed by Pearson correlation and one-way analysis of variance (ANOVA). Duncan’s multiple range test were subjected to analysis mean separations, and differences at *p* < 0.05 were considered to be significant.

## 5. Conclusions

In conclusion, the treatment with different concentrations of MeJA had a significant effect on the phenolic content, while 10 μM MeJA effectively increased the phenolic accumulation of wounded broccoli. Ten-micrometer MeJA treatment significantly induced the phenylpropanoid metabolism pathway and stimulated PAL, C4H, and 4CL enzymes activities of wounded broccoli, then further accelerated the accumulation of the wound-induced rutin, caffeic acid, and cinnamic acid accumulation after 24 h storage, which contributes to the increase of phenolic accumulation and antioxidant activity in wounded broccoli. These results demonstrate that MeJA in combination with wounding stress can induce phenylpropanoid metabolism for the wound-induced phenolic accumulation in wounded broccoli. Therefore, exogenous applications of MeJA are an efficient and promising attempt for broccoli to increase the phenolic content and antioxidant capacity.

## Figures and Tables

**Figure 1 molecules-24-03537-f001:**
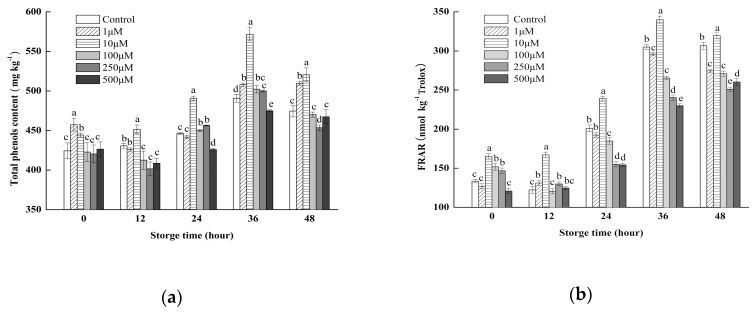
Effect of treatments with different MeJA concentrations on total phenolics content (**a**) and ferric reducing antioxidant power (FRAP) (**b**) of broccoli during 48 h of storage at 20 °C. Columns with vertical bars represent the mean ± SD (*n* = 3).

**Figure 2 molecules-24-03537-f002:**
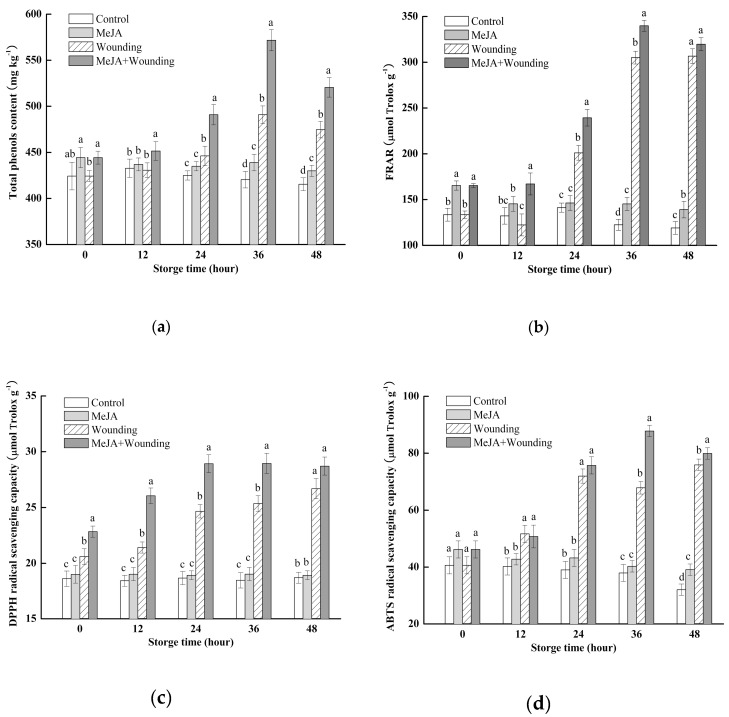
Effect of MeJA treatment and wounding stress on the content of total phenols (**a**), FRAR (**b**), DPPH (**c**), ABTS (**d**) of broccoli during 48 h of storage at 20 °C. Columns with vertical bars represent the mean ± SD (*n* = 3).

**Figure 3 molecules-24-03537-f003:**
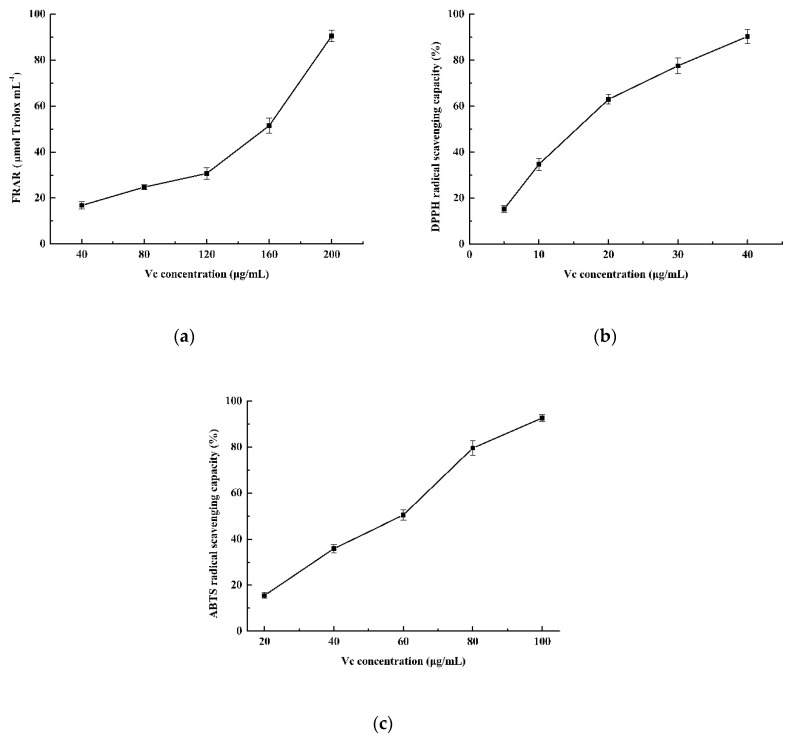
The FRAR (**a**), DPPH(**b**), ABTS (**c**) radical scavenging capacity of the positive control (ascorbic acid) assay.

**Figure 4 molecules-24-03537-f004:**
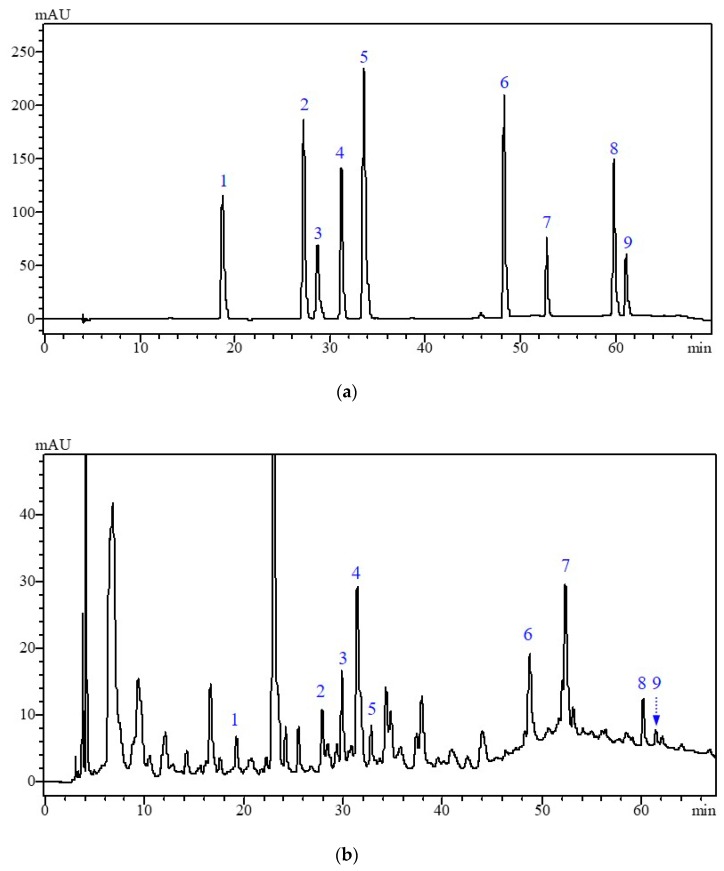
HPLC-PAD chromatograms of phenolic standard (**a**) and individual phenolic compounds from broccoli (**b**) at 280 nm. Peaks 1, 2, 3, 4, 5, 6, 7, 8, and 9 represent catechin, hydroxybenzoic acid, chlorogenic acid, caffeic acid, ferulic acid, sinapic acid, rutin, cinnamic acid, and quercetin, respectively.

**Figure 5 molecules-24-03537-f005:**
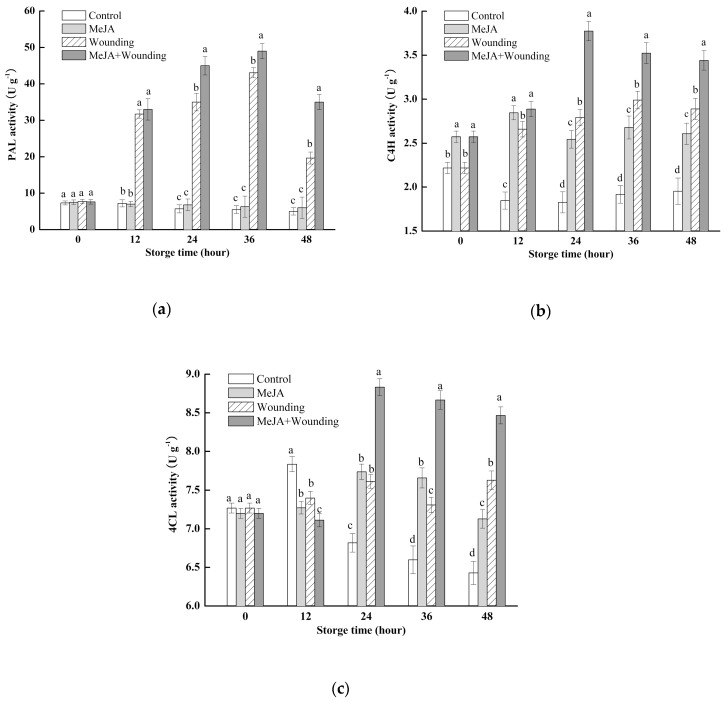
Effect of MeJA treatment and wounding stress on the activities of PAL (**a**), C4H (**b**), 4CL (**c**) of broccoli during 48 h of storage at 20 °C. Columns with vertical bars represent the mean ±SD (*n* = 3).

**Figure 6 molecules-24-03537-f006:**
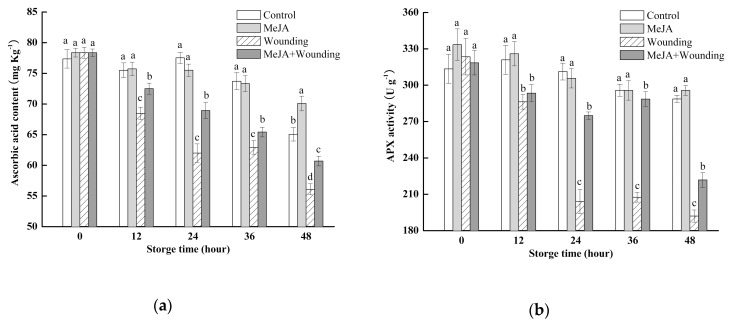
Effect of MeJA treatment and wounding stress on vitamin C content (**a**), APX (**b**), PPO (**c**) and POD (**d**) activities of broccoli during 48 h of storage at 20 °C. Columns with vertical bars represent the mean ±SD (*n* = 3).

**Figure 7 molecules-24-03537-f007:**
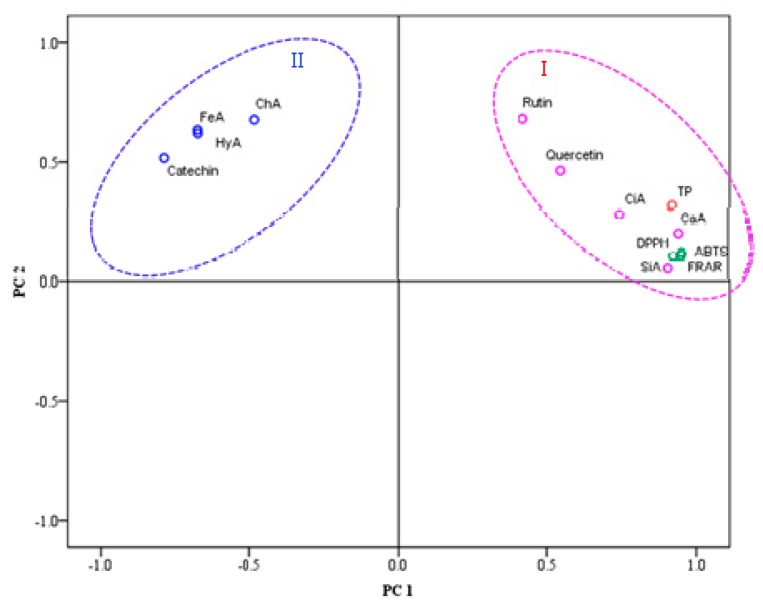
Principal component analysis (PCA) of individual phenols, total phenols, and antioxidant activity of broccoli. Groups Ⅰ and Ⅱ represent the first principal component (PC1) and second principal component (PC2), respectively. CiA represents cinnamic acid. CaA represents caffeic acid. SiA represents sinapic acid. ChA represents chlorogenic acid. FeA represents ferulic acid. HyA represents hydroxybenzoic acid.

**Figure 8 molecules-24-03537-f008:**
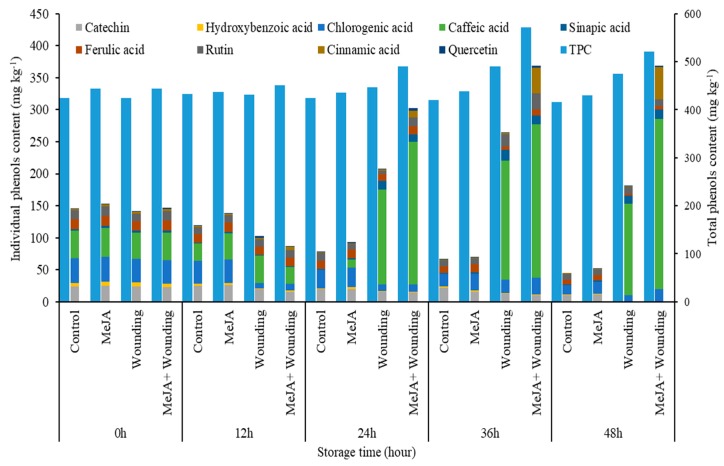
Effect of MeJA treatment and wounding stress on the total amount of individual phenols and TPC of broccoli during 48 h of storage at 20 °C.

**Figure 9 molecules-24-03537-f009:**
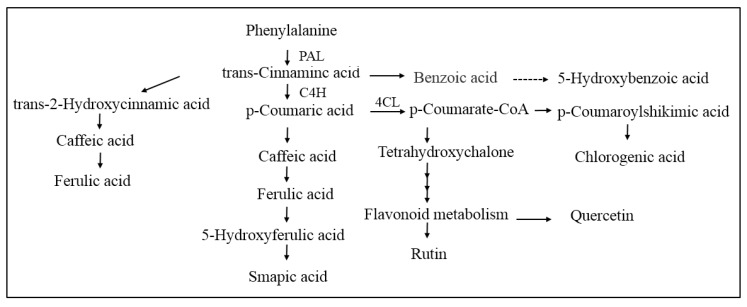
Biosynthesis pathway of phenolic compounds in plants.

**Table 1 molecules-24-03537-t001:** Effect of MeJA treatment and wounding stress on the content of individual phenols of broccoli during 48 h of storage at 20 °C.

Storage (Hour)	Treatment	Individual Phenolic Content (mg kg^−1^)								
		Catechin	Hydroxybenz-oic Acid	Chlorogenic Acid	Caffeic Acid	Sinapic Acid	Ferulic Acid	Rutin	Cinnamic Acid	Quercetin
0	Control	23.87 ± 3.01 ^aA^	5.64 ± 0.45 ^abA^	38.76 ± 2.47 ^aA^	42.59 ± 3.51 ^aA^	2.94 ± 0.19 ^aA^	15.52 ± 1.16 ^aA^	13.02 ± 0.98 ^abA^	2.81 ± 0.21 ^aA^	1.15 ± 0.09 ^aA^
	MeJA	24.92 ± 1.38 ^aA^	6.08 ± 0.58 ^aA^	39.52 ± 2.31 ^aA^	44.50 ± 2.09 ^aA^	3.08 ± 0.21 ^aA^	16.61 ± 0.57 ^aA^	14.37 ± 0.25 ^aA^	2.95 ± 0.31 ^aA^	1.19 ± 0.05 ^aB^
	Wounding	24.52 ± 2.23 ^aA^	5.35 ± 0.62 ^abA^	37.38 ± 1.52 ^aA^	41.33 ± 1.28 ^aC^	2.81 ± 0.25 ^aC^	14.05 ± 0.87 ^aA^	12.15 ± 0.33 ^bB^	2.91 ± 0.17 ^aA^	1.17 ± 0.11 ^aB^
	MeJA + Wounding	22.87 ± 1.28 ^aA^	5.02 ± 0.23 ^bA^	36.67 ± 1.43 ^aA^	43.22 ± 2.06 ^aD^	3.02 ± 0.31 ^aC^	16.33 ± 1.02 ^aA^	15.02 ± 0.66 ^aC^	3.05 ± 0.23 ^aE^	1.25 ± 0.13 ^aE^
12	Control	24.92 ± 2.17 ^aA^	3.05 ± 0.32 ^bB^	36.24 ± 2.93 ^aA^	26.69 ± 1.99 ^bB^	1.59 ± 0.21 ^bB^	13.28 ± 1.21 ^aAB^	10.77 ± 0.99 ^aB^	1.93 ± 0.09 ^cB^	0.81 ± 0.05 ^dB^
	MeJA	25.60 ± 1.93 ^aA^	4.09 ± 0.33 ^aB^	35.99 ± 3.01 ^aA^	40.85 ± 3.01 ^aA^	2.71 ± 0.19 ^aA^	14.09 ± 1.41 ^aB^	11.97 ± 1.01 ^aB^	2.59 ± 0.13 ^bA^	1.14 ± 0.08 ^cB^
	Wounding	19.32 ± 0.96 ^bB^	2.03 ± 0.18 ^cB^	7.87 ± 0.81 ^cD^	43.11 ± 2.94 ^aC^	0.88 ± 0.05 ^cD^	12.93 ± 1.08 ^aAB^	11.62 ± 1.02 ^aBC^	2.02 ± 0.15 ^cB^	2.84 ± 0.14 ^aA^
	MeJA+ Wounding	15.49 ± 0.84 ^cB^	2.59 ± 0.21 ^cB^	10.26 ± 0.96 ^bE^	26.04 ± 1.67 ^bE^	1.34 ± 0.11 ^bD^	13.28 ± 1.09 ^aB^	12.07 ± 1.09 ^aD^	4.49 ± 0.31 ^aD^	1.99 ± 0.13 ^bD^
24	Control	19.35 ± 1.25 ^aB^	1.77 ± 0.12 ^bC^	28.92 ± 1.57 ^aB^	-	1.12 ± 0.09 ^cC^	12.37 ± 1.22 ^aAB^	13.39 ± 1.12 ^aA^	1.02 ± 0.06 ^dC^	0.79 ± 0.06 ^cB^
	MeJA	20.22 ± 2.01 aB	2.44 ± 0.26 ^aC^	30.28 ± 2.45 ^aB^	12.91 ± 1.22 ^cB^	2.08 ± 0.19 ^bB^	13.71 ± 1.09 ^aB^	8.09 ± 0.56 ^bC^	2.03 ± 0.19 ^cB^	1.86 ± 0.12 ^bA^
	Wounding	15.33 ± 1.33 ^bC^	1.32 ± 0.23 ^cC^	10.41 ± 1.02 ^bC^	148.74 ± 9.21 ^bB^	12.68 ± 1.05 ^aB^	10.82 ± 1.03 ^bB^	5.26 ± 0.32 ^cD^	2.58 ± 0.16 ^bA^	0.82 ± 0.07 ^cC^
	MeJA+ Wounding	14.12 ± 1.09 ^bB^	1.03 ± 0.32 ^cC^	12.24 ± 1.01 ^bD^	222.11 ± 8.66 ^aC^	12.39 ± 1.23 ^aB^	12.14 ± 1.05 ^bB^	13.56 ± 1.03 ^aD^	10.23 ± 0.81 ^aC^	4.82 ± 0.37 ^aA^
36	Control	21.75 ± 1.72 ^aB^	2.49 ± 0.48 ^aB^	20.26 ± 1.95 ^bC^	-	0.76 ± 0.05 ^dD^	10.58 ± 0.92 ^bB^	9.29 ± 0.58 ^cB^	0.58 ± 0.02 ^dD^	0.57 ± 0.04 ^cC^
	MeJA	15.32 ± 1.37 ^bC^	2.62 ± 0.24 ^aC^	26.51 ± 1.83 ^aB^	-	1.78 ± 0.12 ^cC^	12.63 ± 1.09 ^aB^	8.82 ± 0.73 ^cC^	1.43 ± 0.07 ^cC^	0.64 ± 0.06 ^cC^
	Wounding	12.56 ± 1.09 ^cD^	1.24 ± 0.11 ^bC^	20.91 ± 1.69 ^bB^	185.33 ± 1.88 ^bA^	17.28 ± 0.97 ^aA^	5.35 ± 0.87 ^cC^	18.71 ± 1.03 ^bA^	2.69 ± 0.21 ^bA^	0.98 ± 0.07 ^bC^
	MeJA+ Wounding	10.28 ± 1.02 ^dC^	0.94 ± 0.05 ^cC^	26.86 ± 2.33 ^aB^	239.48 ± 1.59 ^aB^	13.57 ± 1.24 ^bAB^	9.54 ± 0.64 ^bC^	24.92 ± 1.82 ^aA^	39.42 ± 2.19 ^aB^	3.18 ± 0.28 ^aB^
48	Control	10.08 ± 0.92 ^aC^	0.82 ± 0.04 ^aD^	15.39 ± 1.26 ^bD^	-	0.69 ± 0.05 ^dE^	7.15 ± 0.49 ^bC^	9.47 ± 0.62 ^cB^	0.56 ± 0.04 ^dD^	-
	MeJA	11.87 ± 0.64 ^aD^	0.90 ± 0.06 ^aD^	18.97 ± 1.44 ^aC^	-	1.48 ± 0.91 ^cD^	8.72 ± 0.52 ^aC^	8.67 ± 0.59 ^cC^	0.82 ± 0.06 ^cD^	0.15 ± 0.01 ^cD^
	Wounding	-	-	10.07 ± 0.72 ^cC^	143.49 ± 1.32 ^bB^	12.45 ± 1.06 ^bB^	2.94 ± 0.22 ^dD^	10.50 ± 0.29 ^bC^	1.59 ± 0.13 ^bC^	0.70 ± 0.04 ^bD^
	MeJA+ Wounding	-	-	20.06 ± 1.37 ^aC^	265.28 ± 9.34 ^aA^	15.44 ± 1.09 ^aA^	4.86 ± 0.41 ^cD^	18.62 ± 1.57 ^aB^	50.23 ± 3.64 ^aA^	2.45 ± 0.18 ^aC^

Note: Data are expressed as the mean ±SD (*n* = 3). Values with different letters were significantly different at *p* < 0.05. Lowercase letters represented a significant difference among treatment factors, and capital letters represented a significant difference among storage time factors. Means not detected.

**Table 2 molecules-24-03537-t002:** Pearson correlation coefficients of nine kinds of individual phenols, total phenol (TP) content, PAL, C4H, 4CL, PPO, POD enzyme activities. CiA, CaA, SiA, ChA, FeA, and HyA represent cinnamic acid, caffeic acid, sinapic acid, chlorogenic acid, ferulic acid, and hydroxybenzoic acid, respectively.

	PAL	C4H	4CL	PPO	POD	TP	Catechin	HyA	ChA	CaA	SiA	FeA	Rutin	CiA	Quercetin
**PAL**	1	0.770 **	0.647 **	−0.411	−0.566 **	0.793 **	−0.500 **	−0.516 *	−0.557 **	0.844 **	0.787 **	−0.392	0.453 *	0.557 *	0.697 **
**C4H**		1	0.688 **	−0.636 **	−0.494 *	0.724 **	−0.559 *	−0.400	−0.434	0788 **	0.717 **	−0.343	0.280	0.543 *	0.652 **
**4CL**			1	−0.377	− 0.145	0.769 **	−0.357	−0.312	−0.096	0.776 **	0.642 **	−0.146	0.307	0.671 **	0.761 **
**PPO**				1	0.010	−0.642 **	0.796 **	0.671 **	0.416	−0.585 **	−0.621 **	0.693 **	−0.158	−0.501 *	−0.171
**POD**					1	−0.271	0.099	0.149	0.456	−0.282	−0.208	0.102	−0.254	−0.070	−0.357
**TP**						1	−0.585	−0.456 *	−0.188	0.891 **	0.812 **	−0.447 *	0.656 **	0.824 **	0.595 **
**Catechin**							1	0.839 **	0.650 **	−0.605 **	−0.638 **	0.889 **	−0.011	−0.529 *	−0.129
**HyA**								1	0.775 **	−0.478 *	−0.514 *	0.841 **	0.082	−0.364	−0.163
**ChA**									1	−0.302	−0.325	0.596 **	0.253	−0.058	−0.227
**CaA**										1	0.949 **	−0.511 *	0.436	0.743 **	0.606 **
**SiA**											1	−0.618 **	0.362	0.579 **	0.392
**FeA**												1	0.008	−0.354	0.103
**Rutin**													1	0.410	0.408
**CiA**														1	0.511 *
**Quercetin**															1

Note: * Significant at *p* < 0.05 probability level. ** Significant at *p* < 0.01 probability level.

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
