# Peer review of "Effect of Methyl Jasmonate on Phenolic Accumulation in Wounded Broccoli"

_molecules, 2019, doi:10.3390/molecules24193537_

Round 1

Reviewer 1 Report

In the manuscript, authors report elicitation of broccoli using MeJa in the context of phenolic compound production. The paper contains valuable contents. However, in my opinion, the manuscript in its present form can not be accepted for publication and the manuscript needs major revision before publication in the journal.

Below there are some detailed comments on the paper:

The manuscript does not contain some information needed to evaluate its results. The extraction efficiency was not given, nor was the concentration of the extract used in the tests. For example, line 289- “1 mL of supernatant was added …”  or line 324 “The ethanol extract supernatant (0.1 mL) was added into ..”, but we do not know the concentration of these extracts. Using a given amount of solvent gives different extraction efficiency for different raw materials. In the manuscript,  some results are described unclearly, and some problems are not discussed. Especially, no discussion of the relationship between the individual factors determined - between total polyphenol content, individual compound content and enzyme activities or antioxidant activity. It could be also worth calculating the correlations for all factors analyzed in the experiments. In fragments, the text is difficult to read and it is not known what's going on. Table 1 is very complicated and difficult to read. May be it could be easier, when treatments and elicitation period would be statistically compared, simultaneously.

Why was control (day 0) not included in the statistical analysis?

4. Fig. 2, a comparison should also be made for treatments and theirs times together.

5. Not only the optimal MeJa concentration could be determined based on the preliminary study, but also the elicitation time. Why later all the time points, 12, 24, 36 and 48 hours, were analyzed, it obscures the results by a very lot of described results.

6. The manuscript is chaotically written and contains a lot of mistakes:

For example,

Line 37-38 “… and fresh-cut broccoli has become more popular recently due to its (…) fresh properties” What do the authors mean?

Line 297 - 2,6-dichlorpphenol instead of 2,6-dichlorophenol

Line 297 indophenols solution instead of indophenol solution

Line 305-307: “…0~5 min …” -   “~” it is not “to” , only “more or less”,  “approximately”

Line 334-335: “ …and expressed as nmol·kg-1 (???) Trolox equivalents per kilogram(???) of fresh weight tissue..”

Some mistakes in the bibliography- First names are given as surnames, for example,

- line 442: Fernando, R.L., Emilio, V.J…..

- line 434: Ana, T.; Carolina, S.; Adriana, P.; Mauricio, G.A.; Perla, R.; Luis, C.Z., Daniel.; Alberto, J.V.....

-line 493: Krystian, M.; Łukasz, W.; Francisco, J.B.; Sylwia, S.; Jose, M. L.; Alessandro, Z.; Sara, S....

In Materials an Methods section: “..results were expressed as mean ± SD (standard deviation)”, but in Fig. 1 , 2 –“ Columns with vertical bars represent the mean ± SE” No statistical analysis – Fig. 1 2. Fig 1 a, b – Is there no difference between 12 and 24 days for metabolite contents (1a), but there is large for activities (1b)? I do not agree with some results of the statistics for Table 1, for example, rutin after 12 days: control and MeJA+ Wounding – is it really statistically difference? Statistical analysis needs to be checked. Tab 2. is not needed. It is clear how the different treatments differ without this pattern. Some Figures are colored, and some - black and white - lack of consistency in the presentation of data The conclusions are incorrectly formed: “MeJA treatment significantly stimulated PAL, C4H, and 4CL enzymes activities in phenylpropanoid metabolism..” It is true only with for MeJA together with wounding or “…. further accelerated the accumulation of the wound-induced chlorogenic acid, rutin, caffeic acid and cinnamic acid accumulation..” - This is not true in all cases.

Author Response

Dear reviewer:

We are very grateful to your comments for the manuscript. According to your comments, we amended the relevant part in manuscript. Due to a figure of Principal Component Analysis (PCA) and a table of correlation coefficients were added to the discussion section, use a word file could present the section clearer. For your convenience, please see the attachment, thank you very much. Have a nice day!

Reviewer 2 Report

I have a few minor comments:

Line 18: please check “concentrateion”

Line 36: please be more specific which polyphenols

Line46-48 and 64-68- the sentences must be reformulated

Author Response

Dear reviewer:

Thank you very much for your valuable comments and we have made modifications accordingly. The amendments were highlighted in red color. Have a nice day!

I have a few minor comments:

Response: Thank you for your comments and for your recognition of my work. The manuscript has been revised carefully according to your valuable suggestions. Detailed responses are listed below point by point.

Point 1: Line 18: please check “concentrateion”

Response 1: Thanks for your comment. “concentrateion” has been corrected to “concentration” at line 18.

Point 2: Line 36: please be more specific which polyphenols

Response 2: Thanks for your suggestion. We have added the specific information of polyphenols to the revised paper at line 36-38, the detailed as shown below.

Broccoli is a well known green vegetable with great commercial value owing to its special nutritional and functional components such as phenolic compounds (quercetin, caffeic, quinic acid, chlorogenic acid, etc.) [1,2], vitamin C (Vit C) and other bioactive compounds (Zn, fiber, glucosinolates, etc.).

The added references were as follows:

Yea, J.H.; Huang, L.Y.; Terefe, N.S.; Augustin, M.A. Fermentation-based biotransformation of glucosinolates, phenolics and sugars in retorted broccoli puree by lactic acid bacteria. Food Chem. 2019. 286, 616-623. https://doi.org/10.1016/j.foodchem.2019.02.030 Torres-Contreras, A.M.; Nair, V.; Cisneros-Zevallos, L.; Jacobo-Velázquez, D.A. Stability of Bioactive Compounds in Broccoli as Affected by Cutting Styles and Storage Time. Molecules, 2017. 22(4):636. https://doi.org/10.3390/molecules22040636.

Point 3: Line 46-48 and 64-68- the sentences must be reformulated

Response 3: Thank you for reminding us the improper description on the sentences. We have revised the sentences as shown below.

1) As one of the most important secondary metabolites, phenolic compounds are synthesized through phenylpropanoid metabolic pathway in fresh cut produce. In phenylpropanoid metabolic pathway, PAL, C4H and 4CL are three key enzymes that catalyse the synthesis of phenolic compounds in plants. (line 47-50)

2) Therefore, the phenolic compounds content, the key enzymes that catalyse the synthesis of phenolic compounds and the antioxidant capacity were investigated in wounded broccoli after treatment with MeJA, in order to evaluate the effect of MeJA treatment on phenolic accumulation in wounded broccoli. (line 66-69)

Reviewer 3 Report

Comments to the authors

The study is interesting, quite original and the experimental design is appropriate and the methodology well conducted. The only criticism concerns the discussion section which must be extensively improved in order to better explain the importance of the increase of phenolic compounds and antioxidant activity both for consumers’ health and for the preservation of the product. In this regard it is also necessary to add appropriate bibliographical references.

Minor concerns:

L49: …of carrot. Replace of with in.

L50: …of apple. Replace of with in.

L57: 1μmol. Insert space.

L95: “…correlation coefficient was…”. Replace with “correlation coefficients were…”.

L108: replace compare with compared.

L109: it is not appropriate to start the sentence with while, please edit.

L132: please delete “and”.

L136: “…there is…”. Please use the past tense as in the rest of the paragraph.

L148: “…there is…”. Please use the past tense as in the rest of the paragraph.

L153: “…there is…”. Please use the past tense as in the rest of the paragraph.

L197: 20°C. Insert space.

L261-262: “…then put them in ventilator to dry naturally”. How long?

L267: 20 cm×10 cm×4 cm. Please replace with 20x10x4 cm.

L312: What was the range of the standard curve. Moreover indicate the R2.

L334: nmol·kg-1. In the other cases the point indicating the multiplication has not been inserted. Please stay consistent for the entire manuscript.

L336: 1μM. Insert space.

L346-347: (enzyme activity). Already reported in the line 345, please delete.

L347: =0.01. Insert space.

L354: “…kg-1on…” and “…U=0.01…”. Insert spaces.

L363: 1mM. Insert space.

L367: =0.01. Insert space.

L371: U=0.01. Insert spaces.

L379-380: The sentence is confused. Please revise.

L383-385: Please change as follow, “…which contribute to the increase of phenolic accumulation and antioxidant activity in wounded broccoli”.

L493: Remove bold.

Author Response

Dear reviewer:

I am very grateful to your comments for the manuscript. According to your advices, we amended the relevant parts in manuscript. Some of your questions were answered below. Due to a figure of Principal Component Analysis (PCA) was added to the discussion section, use a word file could present the section clearer. For your convenience, please see the attachment, thank you very much.

Comments and Suggestions for Authors

Comments to the authors

The study is interesting, quite original and the experimental design is appropriate and the methodology well conducted. The only criticism concerns the discussion section which must be extensively improved in order to better explain the importance of the increase of phenolic compounds and antioxidant activity both for consumers’ health and for the preservation of the product. In this regard it is also necessary to add appropriate bibliographical references. 

Response: Thank you for your comments and for your recognition of my work. Detailed responses are listed below point by point.

Minor concerns:

Point 1: L49: …of carrot. Replace of with in.

Response 1: “…of carrot” has been corrected to “…in carrot” at line 51.

Point 2: L50: …of apple. Replace of with in.

Response 2: “…of apple” has been corrected to “…in apple” at line 52.

Point 3: L57: 1μmol. Insert space.

Response 3: 1μmol has been corrected to 1 μmol at line 59.

Point 4: L95: “…correlation coefficient was…”. Replace with “correlation coefficients were…”.

Response 4: “…correlation coefficient was…” has been replace with “correlation coefficients were…”at line 100.

Point 5: L108: replace compare with compared

Response 5: “compare” has been corrected to “compared” at line 113.

Point 6: L109: it is not appropriate to start the sentence with while, please edit.

Response 6: At the beginning of the sentence, the word “while” has been replace with “moreover” at line 113.

Point 7: L132: please delete “and”.

Response 7: “and” has been delete. The sentence “and then experienced a relatively stable trend afterward” has been corrected to “then experienced a relatively stable trend afterward”

Point 8: L136: “…there is…”. Please use the past tense as in the rest of the paragraph.

Response 8: “…there is…” has been corrected to “…there was…” at line 140.

Point 9: L148: “…there is…”. Please use the past tense as in the rest of the paragraph.

Response 9: “…there is…” has been corrected to “…there was…” line 153.

Point 10: L153: “…there is…”. Please use the past tense as in the rest of the paragraph.

Response 10: “…there is…” has been corrected to “…there was…” line 158.

Point 11: L197: 20°C. Insert space.

Response 11: 20°C has been corrected to 20 °C at line 226.

Point 12: L261-262: “…then put them in ventilator to dry naturally”. How long?

Response 12: Dry naturally for 2 h. The sentence “…then put them in ventilator to dry naturally” has been corrected to “…then put them in ventilator to dry naturally for 2 h” line 299.

Point 13: L267: 20 cm×10 cm×4 cm. Please replace with 20x10x4 cm.

Response 13: 20 cm×10 cm×4 cm has been corrected to 20x10x4 cm at 304.

Point 14: L312: What was the range of the standard curve. Moreover, indicate the R2.

Response 14: The standard curves of individual phenolic compounds were added to the revised paper at line 350-355.

The standard curves of catechin, hydroxybenzoic acid, chlorogenic acid, caffeic acid, sinapic acid, ferulic acid, rutin, cinnamic acid and quercetin were y = 15670x + 4058 (R² = 0.9998); y = 43042x + 19721 (R² = 0.9997); y = 41024x + 611.36 (R² = 0.9998); y = 7511.4x + 1641.4 (R² = 0.9965); y = 76156x – 214369 (R² = 0.9629); y = 79743x + 28630 (R² = 0.9997); y = 21598x + 4702.6 (R² = 0.9998); y = 237973x + 75549 (R² = 0.9998); y = 38634x + 2925.6 (R² = 0.9999) respectively, the range of these standard curves was 0.1 to 100 μg mL-1.

Point 15: L334: nmol·kg-1. In the other cases the point indicating the multiplication has not been inserted. Please stay consistent for the entire manuscript.

Response 15: We have got rid of the point of the unit for the entire manuscript.

Point 16: L336: 1μM. Insert space.

Response 16: 1μM has been corrected to 1 μM at line 378.

Point 17: L346-347: (enzyme activity). Already reported in the line 345, please delete.

Response 17: the sentence “where U (enzyme activity) = 0.01 △A290 nm per min” has been corrected to “where U = 0.01 △A290 nm per min” at line 388.

Point 18: L347: =0.01. Insert space.

Response 18: =0.01 has been corrected to = 0.01 at line 388.

Point 19: L354: “…kg-1on…” and “…U=0.01…”. Insert spaces.

Response 19: “…kg-1on…” and “…U=0.01…” have been corrected to “…kg-1 on…” and “…U = 0.01…” at line 396.

Point 20: L363: 1mM. Insert space.

Response 20: 1mM has been corrected to 1 mM at line 405.

Point 21: L367: =0.01. Insert space.

Response 21: =0.01 has been corrected to = 0.01 at line 409.

Point 22: L371: U=0.01. Insert spaces.

Response 22: U=0.01 has been corrected to U = 0.01 at line 413.

Point 23: L379-380: The sentence is confused. Please revise.

Response 23: the sentence means the treatment with different concentrations of MeJA had a significant effect on phenolic accumulation, while 10 μM MeJA effectively increased phenolic accumulation of wounded broccoli. And the sentence “In conclusion, MeJA treatment had a significant effect on phenols content, while 10 μM MeJA effectively increased phenolic accumulation of wounded broccoli” has been corrected to “In conclusion, the treatment with different concentrations of MeJA had a significant effect on phenolic accumulation, while 10 μM MeJA effectively increased phenolic accumulation of wounded broccoli” at line 421-423.

Point 24: L383-385: Please change as follow, “…which contribute to the increase of phenolic accumulation and antioxidant activity in wounded broccoli”.

Response 24: the sentence “…which was contribute to result of phenolic accumulation increasing by 134.8 % and antioxidant activity increasing by 1.3 - 2.5 folds of wounded broccoli.” has been corrected to “…which contribute to the increase of phenolic accumulation and antioxidant activity in wounded broccoli” at line 426-427.

Point 25: L493: Remove bold.

Response 25: the bold was removed and the word “34” has been corrected to “34”.

Point 26: The only criticism concerns the discussion section which must be extensively improved in order to better explain the importance of the increase of phenolic compounds and antioxidant activity both for consumers’ health and for the preservation of the product. In this regard it is also necessary to add appropriate bibliographical references. 

Response 26: Thank you for your comments. The discussion section has been revised carefully according to your valuable suggestions.

1) The revised discussion section marked with red color as shown below.

As the important antioxidant, phenols have as promoters of human health through their scavenging activity by preventing chronic diseases such as cardio-vascular diseases, cancers, type 2 diabetes, neurodegenerative diseases [1,2]. Therefore, it is necessary to find an effective technology that can ensure the delivery of products with high levels of the desired antioxidants. The study showed that the phenols content of wounded broccoli was markedly affected by different MeJA concentrations, and 10 μM MeJA had a significant influence on phenols accumulation (p < 0.05). Similar optimum treatment concentration of MeJA was also found in Chinese bayberries [26]. In other researches, the effective concentration of MeJA on promoting antioxidant capacity ranged from 1-1000 μM [27,28], this result may be caused by different treatment methods and the type of fruits and vegetables. Our studies showed that low concentration (1 μM) MeJA had little effect on TPC in broccoli during all the storage period, which probably because of the concentration of MeJA was too low to induce mechanism of resistance to wounding stress. When the concentration of MeJA greater than 100 μM, MeJA treatment had a slight inhibition effect on TPC in broccoli, this result is in agreement with a previous report, where the application of 250 ppm MeJA to wounded broccoli for 24 h at 20 °C did not induce a significant increase in the concentration of total phenolics [14].

Fruits and vegetables contain diverse phenolic compounds, including cinnamic acid, gallic acid, caffeic acid, chlorogenic acid, catechol, epicatechol, guaiacol and its polymers and esters, and they can also be induced to synthesize by mechanical injury during the processing of fresh-cut fruit and vegetables [28,29], which make great contribution to antioxidant activity and further improve the health properties [3,4]. The content of caffeic and sinapic acid increase by 4.35, and 5.87 times at 36 h, which contribute to the enhancement of antioxidant activity in wounded broccoli. Moreover, the accumulation of phenols could scavenge free reactive oxygen radicals and inhibit membrane lipid peroxidation, thereby increasing the plant's resistance to oxidative damage and further preservation of the product [5]. Sinapic acid is one of the principal precursors of lignin [30]. Therefore, the higher levels of sinapic acid observed after storage of wounded broccoli may be related with the wound-induced activation of the phenylpropanoid metabolism, which is required for the biosynthesis of lignin that in wounded plant tissue serves as a water impermeable barrier that prevent excessive water loss and improve the resistance [30, 31]. In order to illustrate the relationship among individual phenols, total phenols content and antioxidant activity, a Principal Component Analysis (PCA) was performed as shown in Figure. According to the PCA results, there were two principal components (PC) among these parameters. The PC1 mainly includes total phenols, antioxidant activity and five kinds of individual phenols (rutin, quercetin, cinnamic acid, caffeic acid, sinapic acid), this result indicated that rutin, quercetin, cinnamic acid, caffeic acid and sinapic acid were contribute to the enhancement of total phenols content and antioxidant activity of wounded broccoli. Among the five kinds of individual phenols in PC1, cinnamic acid, caffeic acid and sinapic acid were closer to the total phenols content and antioxidant activity in the chart of PCA, it means that these individual phenols have the higher contribution to total phenols content and antioxidant activity. The PC2 mainly includes chlorogenic acid, ferulic acid, hydroxybenzoic acid and catechin, this result suggested that the four kinds of individual phenols had the fewer contribution to the increase of total phenols content and antioxidant activity. 

2) The added references  as shown below.

1 Costa, C.; Tsatsakis, A.; Mamoulakis, C.; Teodoro, M.; Briguglio, G.; Caruso, E. Current evidence on the effect of dietary polyphenols intake on chronic diseases. Food chem toxicol. 2017, 110, 286-299. http://dx.doi.org/10.1016/j.fct.2017.10.023. (marked red color at line 515 in the revised paper)

2 Perez-Jimenez, J.; Neveu, V.; Vos, N.; Scalbert, A. Systematic analysis of the content of 502 polyphenols in 452 foods and beverages: an application of the phenol-explorer database. J. Sci. Food Agr. 2010, 58(8), 4959-4969. https://doi.org/10.1021/jf100128b. (marked red color at line 518 in the revised paper)

3 Kenny, O.; O’Beirne, D. Antioxidant phytochemicals in fresh-cut carrot disks as affected by peeling method. Postharvest Biol. Technol. 2010, 58, 247–253. https://doi.org/10.1016/j.postharvbio.2010.07.012. (marked red color at line 533 in the revised paper)

4 Heo, H.J.; Kim, Y.J.; Chung, D.; Kim, D.O. Antioxidant capacities of individual and combined phenolics in a model system. Food Chem. 2007, 104 (1), 87–92. https://doi.org/10.1016/j.foodchem.2006.11.002. (marked red color at line 535 in the revised paper)

5 Papuc, C.; Goran, G.V.; Predescu, C.N.; Nicorescu, V.; Stefan, G. Plant polyphenols as antioxidant and antibacterial agents for shelf‐life extension of meat and meat products: classification, structures, sources, and action mechanisms. Compr. Rev. Food Sci. F. 2017, 16(1), 1243–1268. https://doi.org/info:doi/10.1111/1541-4337.12298. (marked red color at line 537 in the revised paper)

Round 2

Reviewer 1 Report

The manuscript has undergone radical improvement, errors have also been corrected. Essential parts have been developed and supplemented.
It would be worth supplementing the manuscript with the results of antioxidant properties (FRAP, ABTS, DPPH) obtained for known antioxidants (such as ascorbic acid, BHT etc.), because there is no positive control performed in the same conditions in these analyzes.

Author Response

Dear reviewer:

    Thank you very much for your comments and recognition of our work. According to your suggestion, we used ascorbic acid as the positive control of the antioxidant properties (FRAP, ABTS, DPPH) analyzes. And the method of positive control experiment was added to the revised paper (Line 380-382), and the result of the positive control experiment (figure 3) was added to the revised paper at Line 104-108.

    In addition, the method of antioxidant capacity determination of broccoli in our study was according to the previous study reported by Chen [1], Ozgen [2] and Lou [3], the references as shown below.

[1] Chen C.; Hu W.; Zhang R.; Jiang A.; Zou Y. Levels of phenolic compounds, antioxidant capacity, and microbial counts of fresh-cut onions after treatment with a combination of nisin and citric acid. Hortic. Environ. Biotechnol. 2016, 57(3): 266-273. https://doi.org/10.1007/s13580-016-0032-x.

[2] Ozgen M.; Reese, R N.; Tulio A Z.; Scheerens J C.; Miller A R. Modified 2,2-azino-bis-3-ethylbenzothiazoline-6-sulfonic acid (ABTS) method to measure antioxidant capacity of selected small fruits and comparison to ferric reducing antioxidant power (FRAR) and 2,2′-diphenyl-1-picrylhydrazyl (DPPH) methods. J. Agr. Food. Chem. 2006, 54(4), 1151-1157. https://doi.org/10.1021/jf051960d.

[3] Lou H.; Hu Y.; Zhang L.; Sun P.; Lu H. Nondestructive evaluation of the changes of total flavonoid, total phenols, ABTS and DPPH radical scavenging activities, and sugars during mulberry (Morus alba L.) fruits development by chlorophyll fluorescence and RGB intensity values. LWT - Food Sci. Technol. 2012, 47(1):0-24. https://doi.org/10.1016/j.lwt.2012.01.008.

    Your careful work and thoughtful suggestions helped us to improve the quality of this manuscript substantially. Thank you very much! Have a nice day!